# Empowering Women, Enhancing Health: The Role of Education in Water, Sanitation, and Hygiene (WaSH) and Child Health Outcomes

**DOI:** 10.3390/ijerph22050706

**Published:** 2025-04-29

**Authors:** Aminata Kilungo, Mark Bayer, Zoe Baccam, Hamisi Malebo, Halima Alaofe

**Affiliations:** 1Community, Environment and Policy Department, Mel and Enid Zuckerman College of Public Health, University of Arizona, 1295 N. Martin Ave, Tucson, AZ 85721, USA; 2Epidemiology and Biostatistics Department, Mel and Enid Zuckerman College of Public Health, University of Arizona, 1295 N. Martin Ave, Tucson, AZ 85721, USA; mbayer@arizona.edu (M.B.); zoebaccam@arizona.edu (Z.B.); 3National Commission for UNESCO of the United Republic of Tanzania, Dar-es-Salaam P.O. Box 20384, Tanzania; hmalebo@gmail.com; 4Health Promotion Sciences Department, Mel and Enid Zuckerman College of Public Health, University of Arizona, 1295 N. Martin Ave, Tucson, AZ 85721, USA; halaofe@arizona.edu

**Keywords:** WaSH, socio-determinants of health, education, diarrhea, Tanzania

## Abstract

**Background:** Adequate water, sanitation, and hygiene (WaSH) are critical to maintaining good health and hygiene. However, health is a function of many health determinants, and WASH services alone may not be sufficient to improve health outcomes. **Objective:** To identify whether the presence of WaSH services is associated with fewer children under five years of age experiencing symptoms of diarrhea in Katoma, Geita, Tanzania. **Method:** A cross-sectional study was conducted to collect health data, demographics, and other variables, such as WASH, food insecurity, education of the mother, vaccination data, and household income data, for 452 households with children under five. Surveys were completed in-person through interviews. Health outcome data included being sick with diarrhea or symptoms. Data analysis was performed using SAS OnDemand for Academics. Multivariate logistic regression and mixed-effects logistic regression models were employed to determine the association between the covariates and sickness of inclusion children and all the children involved in the study, respectively. **Results:** The findings suggest that WASH services alone do not have a significant impact on diarrhea, but other determinants of health, including the education of the mother, showed a significant impact on health outcomes among children with at least one WASH service. These demographic variables were also associated with lower food insecurity and poverty. The findings highlight the need to (1) include other covariates when analyzing WASH data to understand health outcomes; and (2) improve education attainment for women to maximize health benefits for their children.

## 1. Introduction

Water, sanitation, and hygiene (WaSH) are critical factors impacting human health and well-being. Despite this fact, in 2020, 771 million people lacked essential water services, including 122 million still drinking surface water [1]. Furthermore, an estimated 1.7 billion people did not have adequate sanitation services, while 2.3 billion lacked basic hygiene services [1] (WHO, 2021). For women and children, WaSH services have a much higher impact on their health, dignity, and socio-economic development. Women need additional services relating to menstrual health, such as a private place to clean themselves, change, and utilize menstrual products. Moreover, when menstrual services are not present, this affects their dignity and creates disparities between men and women in the ability to participate in social and economic activities (school, work, etc.). Water collection can also take a significant amount of time, excluding women from fully participating in economic opportunities, and thus further widening WaSH-related gender disparities [1,2]. For children, inadequate WaSH services may contribute to diarrhea diseases, and thus impact their health and growth [3]. As such, lack of access to adequate WaSH services goes beyond direct health benefits. Even though the benefits of WASH are well documented, gaps still exist in the socio-determinants of WaSH at the community level. Socio-determinants of WaSH are the conditions in the environments where people are born, live, learn, work, play, worship, and age that affect a wide range of health, functioning, and quality-of-life outcomes and risks [4]. For instance, WASH intervention can bring about a reduction in diarrhea disease [5]. Still, there are mixed conclusions from many studies aiming to show the effectiveness of WASH interventions and their health outcomes [6].

WaSH services are also important in the prevention of water-related infectious diseases (WRIDs). WRIDs can be classified as waterborne infections, water-washed infections, water-based infections, and infections associated with water-related insects [7]. These diseases include cholera, typhoid fever, trachoma, and diarrhea. Diarrhea is a water-washed disease, meaning a “disease that is transmitted due to poor hygiene from not having access to safe water” (Center for Disease Control and Prevention [8]). Leading causes of diarrhea include the pathogens rotavirus and norovirus. Diarrhea is especially harmful, as it is a leading cause of death in the world for children aged five years or younger, being responsible for approximately 484,000 deaths each year [9,10]. Over 50% of these deaths related to diarrhea can be attributed to poor WaSH services. Most deaths due to diarrhea occur in LMICs, with up to 90% of global diarrheal deaths occurring in Sub-Saharan Africa (SSA) and South Asia [11].

SSA faces unique challenges in WaSH access, due to its increasing population and historical lack of access to protected WaSH services [12]. In many regions of SSA, urban areas have higher access to WaSH services, yet there are many disparities within these areas. Research shows that urban areas with lower access to other services, such as education or income-generating activities, have less WaSH access [12]. In SSA, lack of safe WaSH resources is a large risk factor for diarrheal diseases, especially in children, and contributes to many adverse health outcomes. In Tanzania, it is estimated that the government spends 70% of its health budget on managing preventable diseases that could be addressed effectively through increased WaSH access and resources [13]. Understanding the impact of WaSH services on health outcomes is essential in continuing to address these diseases and inform public health policy.

Many other factors besides WaSH contribute to the health of children under five, especially in LMICs. Some previously identified risk factors include households in poverty or low-wealth quintiles, having a mother with only a primary level of education, food insecurity, and low vaccination rates [14,15]. This study aimed to determine whether household WaSH access, adjusting for other risk factors, including education, food insecurity and vaccination rates, impacts the incidence of diarrhea or diarrhea symptoms in children under five. We hypothesized that WASH status would be the primary indicator of children’s health in a household.

## 2. Methods

### 2.1. Study Design, Setting, and Participants

A survey was conducted in Katoma, Tanzania, in 2018, through collaboration between the University of Arizona (UA) and the National Institute of Medical Research (NIMR), with approval from the Institutional Review Board (IRB) from both entities (UA IRB 1806669871). Katoma is a village located in the northwest region of the country. The region is considered a low-resource area, which makes it an ideal location to assess the relationship between WaSH services and child health outcomes (Figure 1). A total of 452 homes were surveyed during June and July. The inclusion criteria were at least one child aged five or younger living in the household. Participants were excluded if they did not have a child aged five or younger. A 16-section survey was conducted through a structured interview with the head of the household, who answered for all the household members. Verbal consent was retrieved from each participant. A household was defined as a group of people living in the same dwelling, sharing the same meal, and working on at least one common land area or on one same income-generating activity (e.g., livestock farming, commerce, fishing) under the direction of one person who was the head of household. Data on 2757 individuals were collected to learn about the well-being of households in Katoma, including information about agriculture, food security, WaSH services, income, education, and maternal/child health.

This study focused on the health outcomes of children aged five or younger; therefore, the final dataset used for analysis included children of interest who could be confidently identified (N = 719). Two methods were used to isolate these children from the dataset: by identifying the child as the one who qualified the household for the survey (inclusion child) or confirming the child’s age to be five years or younger (Figure 2).

### 2.2. Outcome

The primary outcome, sickness, was defined by whether or not the child had diarrhea or diarrhea symptoms. It was reported by the head of the household whether the child had experienced diarrhea (with or without blood) or a fever in the 30 days prior to when the interview took place. If a child suffered from diarrhea or a fever, they were labeled as sick. If a child did not suffer from either of these, they were labeled as healthy.

### 2.3. Covariates

Based on previously identified risk factors in the literature, the following indicators were measured: water, sanitation, hygiene, food insecurity, vaccination status, education of the mother, and household income [14,15]. Water was defined according to the safety of the water source and access to the water source. To meet basic water service requirements, water must be from an improved source and water collection must take less than a 30 min round trip [17]. Water was made a binary variable (has basic services, does not have basic services). Sanitation was defined according to the type of toilet the household owned and how many households used the toilet. To meet basic sanitation service requirements, a household must own an improved sanitation facility which is not shared with other households [18]. Sanitation was made a binary variable (has basic services, does not have basic services). Hygiene was defined according to the presence of a handwashing station and the availability of soap and water. To meet basic hygiene service requirements, a household must have a handwashing facility with soap and water present [19]. Hygiene was made a binary variable (has basic services, does not have basic services). WaSH was defined according to the summation of basic services for water, sanitation, and hygiene. WaSH was made a binary variable (no basic services, at least one basic service). Food insecurity, or the lack of regular access to nutritious food, is categorized by the Food and Agriculture Organization of the United Nations. The level of food insecurity for each household was determined according to the variety, size, and number of meals: mild food insecurity is defined as worries about not having enough food, moderate food insecurity is defined as reduced quality and quantity of food, and severe food insecurity is defined as when a person has gone a whole day and night without eating. Food insecurity was made a binary variable (mild or no food insecurity, moderate or severe food insecurity) [20]. Vaccination status was defined according to whether the inclusion child had a vaccination card or not, regardless of whether the vaccination card was shown. Immunity information was available only for the inclusion children, so all children within the same household were given the same vaccination status as the inclusion child. Vaccination status was made a binary variable (yes, no). The education level of the mother was defined according to the last educational class the mother of the inclusion child had completed. The level of education of the mother was made a binary variable (completed primary education, did not complete primary education). Income was defined by the selling of animal products and common property resources, such as eggs, milk, grasses, and crops. The level of income was determined by the usage of these products. Income was made a binary variable (income-generating, sustenance). Hamlet was defined as a categorical variable to represent which hamlet, or small community, the household was located in. Five hamlets were included in this survey.

### 2.4. Missingness

The sickness outcome was considered missing when this response was missing for the child.

All covariates, except for WaSH, were considered missing when any of the responses that determined the variable level were missing for the child. For example, the level of water was based on two factors: water source and water distance. If either response to these questions was missing, then water was considered missing. Missingness for WaSH was handled differently based on its component variables (water, sanitation, and hygiene) (Figure 3). When all non-missing components were “does not have basic services”, then any components for WaSH would be missing. When any non-missing components were “has basic services”, the WaSH status already fell into the “at least one basic service” level; therefore the rest of the components, missing or not, would not change the variable level. Examples and clarification of how WaSH was categorized can be found in Figure 3.

### 2.5. Statistical Methods

Data cleaning was performed in SAS OnDemand for Academics (https://www.sas.com/en_us/software/on-demand-for-academics.html, accessed on 23 April 2025). All data were deidentified before data cleaning and analyses. All analyses were performed in R 4.2.1. Summary statistics (counts and proportions) were derived for all 719 children stratified by WaSH status (Table 1).

Logistic regression and mixed-effects models were used for the primary analyses to determine whether having at least one basic WaSH service reduced the odds of having diarrhea or diarrhea symptoms. The logistic model was used for analyses that only considered the inclusion child, while the mixed model, with a household random effect to allow for clustering or correlated responses within a household, was used for analyses that included all children above 5 years old. Multicollinearity among the covariates was assessed using variance inflation factors (VIFs) to ensure that the covariates were not highly correlated with each other (Appendix A). A VIF value under 5 for all variables means that the presence of multicollinearity is low enough to determine the individual effects of each covariate on the health outcome [21]. The VIFs for mixed-effects models were calculated through eigenvalues of the covariance matrix [22]. Linearity of the log-odds was not assessed, because all the covariates were categorical. Goodness-of-fit for the models was assessed through the Akaike information criterion (AIC), with a lower AIC value indicating a better fit of the data [23].

Multivariable logistic regression models were used to identify whether an association existed between the covariates and sickness for the inclusion child (n = 348). The first was the null model (Model I), which only looked at WaSH as a predictor, without adjusting for any other explanatory factors. The second model (Model II) included the household-level covariates. A hamlet sampling weight was used in both models and assigned to each child, representing the inverse of the probability that the child was from that hamlet [24]. This sampling weight was included due to unequal sampling caused by the hamlets Kisoji and Ruchiri having low representation in the study.

Mixed-effects logistic regression models were used with the same sampling weights to investigate the association between the covariates and sickness for all the children in the study (N = 719). The first mixed-effects model (Model III) only considered a WaSH effect and the household random effect. The second mixed-effects model (Model IV) adjusted for other covariates.

To handle data missingness, multiple imputation by chained equations (MICE) was used. The data met the assumption of missing at random (MAR), supporting the appropriateness of this method. Missingness can be explained by the data entry in the Research Electronic Data Capture (REDCap) website platform. We treated the data as MAR, as most cases of missing data were due to data entry errors. Using analysis of variance, the complete case analyses were compared against the analyses using the imputed MICE datasets. Using MICE, results were not significantly different from the full case analyses. Therefore, the results from the complete case analyses were used going forward.

Despite growing rates of agriculture in Africa, many homes utilize sustenance farming, where families consume most of the farm products they produce for nutrition, instead of for generating income [25,26]. While Katoma also experiences high rates of agriculture, with 429 (94.91%) households in this study having plots used for growing crops in the past year, a large percentage of households consume the produce, leaving none for selling to generate household income (Table 1). Due to this, income, defined by the selling of animal and agricultural products, is not a good indicator of socio-economic status (SES). Asset wealth is a more reliable indicator of SES, especially in low-income areas [27]. Asset wealth was defined as the summation of all assets (household, agricultural equipment, or other), where the top 20% of households were considered “Rich”, the next 40% were considered “Middle”, and the bottom 40% were considered “Poor”.

About 368 households were included in the final dataset, but only 14 households met two basic WaSH service requirements, and none met all three. To investigate whether the children with two basic WaSH services were significantly different from the children with one or zero, WaSH was recoded (or redefined) to represent the number of basic services that child had (0, 1, or 2), based on the number of basic services for water, sanitation, and hygiene. Model II and Model IV were run again. As the children with two basic services had low representation, additional sampling weights were needed. Using the same inverse probability method applied to the hamlets, a weight was assigned to each child to represent the inverse probability of the child having that many basic WaSH services. This WaSH weight was multiplied with the hamlet sampling weight to obtain the final sampling weight that was used when running these models again.

## 3. Results

### 3.1. Study Participants

The results presented in Table 1 provide a comprehensive overview of the distribution of children within the study, stratified by their WaSH status. To ensure the statistical integrity of the analysis, the challenge of small cell counts arising from the limited number of children with two services (n = 25) was addressed by consolidating them with the children who had one service. Notably, the percentage of children who were sick (reporting diarrhea symptoms) exhibited little variation across the WaSH strata, 22.67% and 20.59%. Conversely, intriguing patterns emerged when examining other covariates. A higher percentage of mothers who completed primary education was observed among children with at least one WaSH service (49.02% vs. 44.15%). Additionally, we found that children with at least one basic WaSH service demonstrated a significantly lower percentage of experiencing moderate or severe food insecurity in the past month (31.86% vs. 51.07%). Notably, only 55 children (7.65%) experienced severe food insecurity, indicating that most food insecurity is moderate. Interestingly, more children without basic WaSH services sold crops or animal products for income (15.27% vs. 10.29%). Regarding asset wealth, children with no basic services exhibited the highest percentage in the poor level (48.21%) and only 13.60% in the rich level. In contrast, children with at least one basic WaSH service displayed a lower percentage in the poor level (19.12%), with the middle level being the most prevalent (44.12%). Examining sex distribution, we found an almost equal split for both WaSH strata, indicating minimal gender-based disparities within the study population. Furthermore, the distribution of hamlets was similar between the WaSH strata, with Chang’ombe showing the highest percentage for both strata (34.13% and 36.27%), while Ruchiri exhibited the lowest (5.25% and 4.41%).

### 3.2. Exposure to Inadequate WaSH

Table 2 highlights the odds ratios (ORs), 95% confidence intervals (CIs), and associated *p*-values for the household covariates in Models I–IV. In Model I (n = 254), with WaSH as the only predictor, inclusion children who had at least one basic WaSH service were significantly more likely to be healthy (reporting not having diarrhea symptoms) compared to children who had no basic WaSH services (OR = 2.04, *p*-value = 0.036). When the other covariates were adjusted for in Model II (n = 239), children with at least one basic WaSH service no longer had significantly different odds of being healthy (OR = 1.34, *p*-value = 0.397). The only significant effect was for the Ruchiri hamlet, where the odds of being healthy were 93% lower for inclusion children living there than inclusion children living in Katoma Center (OR = 0.07, *p*-value = 0.012). The odds of being healthy for the non-reference hamlets were not significantly different than for Katoma Center. Still, Mataho and Chang’ombe had strong trends (ORs = 0.41 and 0.51, *p*-value = 0.053 and 0.084, respectively). Another strong trend involved children whose mothers completed at least primary education. These children were 72% more likely to be healthy than children whose mothers did not complete primary education (OR = 1.72, *p*-value = 0.095). The remaining factors of food insecurity, income, and vaccination status were not significant.

### 3.3. Incidence of Diarrhea

For the mixed-effects models including all children with the random effect for the household, Model III (n = 517) only had WaSH as a predictor of children being healthy. The odds of being healthy were not significantly different in children with at least one basic WaSH service compared to those without (OR = 1.57, *p*-value = 0.090). When the other covariates were adjusted for in Model IV (N = 476), WaSH was still not a significant factor influencing the odds of a child having diarrhea or not (OR = 1.19, 0.558). The only significant result was obtained, again, for the Ruchiri hamlet, where the odds of healthy children were 91% lower than those for the Katoma Center hamlet (OR = 0.09, *p*-value = 0.001). Mataho and Chang’ombe, again, had strong trends, but these were not significant (OR = 0.51 and 0.60, *p*-value = 0.053 and 0.088, respectively). Other strong trends included the mother’s education (OR = 1.54, *p*-value = 0.089) and vaccination status (OR = 0.46, *p*-value = 0.063). The remaining factors of food insecurity and income were not significant.

Table 3 gives summaries for Models II and IV from Table 2, but with asset wealth replacing income. None of the family or individual effects are statistically significant at the 5% level, but significant differences are seen between the hamlets Mataho (*p*-value = 0.041 and 0.049 for Model I and IV, respectively) and Ruchiri (*p*-value = 0.026 and 0.001 for Model I and IV, respectively) in Katoma Center.

### 3.4. Associations Between Inadequate WaSH and Diarrhea

Children who had at least one basic WaSH service did not have significantly different odds of being healthy compared to children who did not have one basic WaSH service for both Models II and IV (OR = 1.46 and 1.13, *p*-values = 0.306 and 0.696, respectively). Children in two hamlets, Mataho and Ruchiri, had significantly different odds of being healthy than children in Katoma Center for both models. In Model II, the odds of being healthy were 62% lower and 94% lower for children in Mataho and Ruchiri compared to children in Katoma Center, respectively (OR = 0.38 and 0.06, *p*-values = 0.041 and 0.026). In Model IV, the odds of being healthy were 50% lower and 92% lower in Mataho and Ruchiri, respectively (OR = 0.50 and 0.08, *p*-values = 0.049 and 0.001). Vaccination status in Model IV showed a strong trend regarding the odds of being healthy for children with a vaccination card (OR = 0.50, *p*-value = 0.089). In contrast, the effect of vaccination status in Model II was insignificant (OR = 0.70, *p*-value = 0.398). The remaining factors, education of the mother and food insecurity, were not significant for both models. The AIC value for Model II was 255.08, while the AIC value for Model IV was 451.04.

Table 4 shows Models II and IV from Table 2. Again, WaSH was redefined to represent the number of basic services the child had access to, and a new sampling weight was used for the low representation of children with two basic services. In Model II (n = 239), both children who had at least one basic service and children who had at least two basic services were not statistically different from children with no basic services in the odds of being healthy (OR = 1.21 and 1.53, *p*-values = 0.631 and 0.493, respectively). The same insignificant results were observed in Model IV (n = 476) (OR = 0.94 and 2.52, *p*-values = 0.850 and 0.152, respectively).

### 3.5. Significant Predictors of Diarrhea

Food insecurity and education of the mother in Model II showed significant differences in their influence on the odds of children being healthy. For food insecurity, the odds of being healthy for children with mild-to-no food insecurity were 2.42 times greater than the odds for children with moderate-to-severe food insecurity (OR = 2.42, *p*-value = 0.042). For the education of the mother, the odds of being healthy for children whose mothers had completed primary education were 3.26 times greater than the odds for children whose mothers had not completed primary education (OR = 3.26, *p*-value = 0.013). The remaining factors of vaccination status, income, and hamlet were not significant.

Education of the mother and vaccination status in Model IV showed significant differences in the odds of children being healthy. For the mother’s education, the odds of being healthy for children whose mothers completed primary education were 2.24 times greater than the odds for children whose mothers did not complete primary education (OR = 2.24, *p*-value = 0.027). For vaccination status, the odds of being healthy for children who had a vaccination card were 70% less than that for children who did not have a vaccination card (OR = 0.30, *p*-value = 0.036). Although not statistically significant, food insecurity had a strong trend in its influence on the odds of being healthy for children with mild-to-no food insecurity (OR = 1.99, *p*-value = 0.060). The Ruchiri hamlet also had a strong trend, but this was not significantly different from the Katoma Center hamlet’s odds of healthy children (OR = 0.23, *p*-value = 0.085). The trends for the remaining hamlets and factors of income were not significant.

Table 4 shows Models II and IV from Table 2, but WaSH was redefined to represent the number of basic services the child had access to, and a new sampling weight was used for the low representation of children with two basic services. In Model II (n = 239), both children who had at least one basic service and those who had at least two basic services were not statistically different from children with no basic services in their odds of being healthy (OR = 1.21 and 1.53, *p*-values = 0.631 and 0.493, respectively). The same insignificant results were observed in Model IV (n = 476) (OR = 0.94, and 2.52, *p*-values = 0.850 and 0.152, respectively). Both models showed no differences.

## 4. Discussion

Models II and IV from Table 4 provide the best-fitting models. Therefore, focusing on the results from Table 4, the primary indicator of interest, WaSH, does not significantly differ in the odds of children being healthy when comparing the WaSH strata.

The results show that WaSH alone may not be sufficient to address health outcomes. Children with at least one basic service had a higher odds ratio of being healthy, but when adjusting for other covariates, the mother’s education played a much more significant role in children with at least one basic service being more nutritious, wealthier, and having lower food insecurity. Focusing on other social determinants of health, such as education for the mother, significantly improves health status, and should be considered for health interventions. A higher percentage of mothers who completed primary education was also observed among children with at least one WaSH service. These children also experienced lower food insecurity and a lower percentage of poverty. All four models showed that the education of the mother is the best predictor of reduced risk of diarrhea.

Vaccination status had a significant influence on the odds of children being healthy. However, vaccination status was a more important risk factor compared to results reported in prior studies ([14,15,28]). Diving specifically into rotavirus vaccine research, vaccination appears to reduce child hospitalizations and mortality significantly, but little is known in regard to the reduction of diarrheal episodes [28,29]. In the scope of this study, none of the vaccines listed were for the rotavirus, so if a child had a vaccination card, there was no way of knowing if that child had received the rotavirus vaccine and had any protection against this pathogen. Rotarix, the rotavirus vaccine, was introduced to parts of Tanzania in 2013, through an initiative focused on lowering rotavirus-related hospitalizations [30]. While vaccinations provide significant reductions in rotavirus hospitalizations and mortality, further research needs to be conducted on mild or moderate cases of rotavirus, and the effect that vaccinations have on symptom reduction, such as diarrhea, based on the results from this study.

Food security and education of the mother significantly improved the odds of children being healthy. Food insecurity was used as a proxy variable for nutritional status in this study, given its difficulty to measure. These results align with the literature, as malnutrition is linked with diarrhea, and mothers having higher levels of education is associated with lower odds of their children having diarrhea [15,31]. Previous research has found that this association extends to mothers with secondary education or higher, but we were not able to investigate this, as only six mothers had completed at least secondary education. Prior research also suggests that the education level of the mother is not a sufficient indicator on its own, as having a higher education level forms a relationship with better awareness of health education. Health education can affect several things, such as nutritional practices and adherence to sanitation and hygiene guidelines [32].

To test the overall robustness of the dataset, the proxy variable for SES was changed from income of agricultural products to asset wealth. This change led to no significant differences between the ORs of the common variables. Furthermore, income and wealth resulted in insignificant differences in the odds of children being healthy within their strata. Looking at the AIC values for how well the models fitted the data, Model II with income had an AIC of 271.79, and Model II with wealth had an AIC of 255.08. Model IV with income had an AIC of 482.43, and Model IV with wealth had an AIC of 451.04. Both Models II and IV which included wealth provided a better fit of the data. When comparing the Model IIs and the Model IVs, based on the estimates and AIC values, these models were similar in how well they fitted the data. The models that included wealth did have a lower AIC value, which indicates that wealth may have be a better indicator of SES, but SES on its own is not enough to significantly impact the odds of children having diarrhea or not.

For another test of the overall robustness, WaSH was changed from a binary variable (no services vs. at least one service) to one with three categories indicating the number of confirmed WaSH services (no services vs. at least one service vs. at least two services). A different sampling weight was used to account for the lower representation of children with at least two services. Despite this change, all WaSH outcomes remained insignificant. Food insecurity, education of the mother, and vaccination status all became significant factors with the changes in WaSH and sampling weight, while the hamlet Ruchiri lost significance. The other hamlets remained insignificant when comparing Model II and Model IV, but it is interesting to note the change in direction of the OR estimates. The lower AIC values for Model II and Model IV, of 130.16 and 288.45, after these changes indicate that executing these changes provided models that fitted the data better, compared to the original models in Table 2. Considering the numerous changes to variables and large improvements in the AIC values, these models provided a better fit to the data.

## 5. Conclusions

Diarrhea continues to be a major problem in LMICs, including Tanzania. In this study, factors were identified at the household level that were associated with diarrhea in the past month among children under five years old. The highest odds of being free from diarrhea were among children who had mild or no food insecurity and who had mothers who had completed at least primary education. The number of WaSH services that the child had access to was not found to significantly impact the odds of having diarrhea. These results suggest that other socio-determinants of health must be considered to improve health outcomes. Interventions focusing on WaSH alone may not yield the desired health outcomes.

### 5.1. Strengths and Limitations

There are various strengths to this study, including the large sample size across five different hamlets in Katoma, Tanzania, and the in-depth survey format that collected data on multiple topics beyond the scope of this study.

However, the study was also subject to several limitations. Firstly, the mother interviewed was the only mother/caregiver that could be confidently identified. Due to this, their education level was attributed to all children under five years old within the household. The households in this dataset had up to 19 people; therefore, it is very likely that in several households, there were multiple mothers/caregivers with varying levels of education, but only one was assessed. Similarly, all children within a household inherited the vaccination status of the inclusion child. This may have provided inaccurate data regarding the vaccination status of the children. A second limitation is that income was very limited. Questions were limited to the selling of animal products and common property resources as the only source of income. Some records did not have complete data, leading to high missingness for some variables, including 15% of the outcome variable. This could potentially impact the strength of some of the findings. The study also relied on self-reported data, which may mean that the results are susceptible to recall bias. Along similar lines, vaccination status was not confirmed by a vaccination card, but relied on self-reporting. The technology difficulties that caused the study team to enter data by hand from paper surveys may also have influenced the high level of missingness. Finally, although Models II and IV from Table 4 had the best fits to the data, it should be noted that the 95% confidence intervals were much wider in comparison to Table 2 or Table 3, indicating possible instability in the point estimate ORs.

### 5.2. Future Research

Future research should prioritize understanding how multiple covariates and disparities impact the health outcomes of children under 5 in combination with WaSH status and access. Future research should aim to understand how variables such as the mother’s education impacts the way in which WaSH is communicated to children, and how mothers are taught to use WaSH to improve the health of their children. Our findings also highlight the need to integrate WASH into other community health interventions, such as improving nutrition, to improve health outcomes [33]. WASH interventions alone may not provide the desired outcome. More studies are needed to show the co-benefits of integrating WASH into other community health programs.

### 5.3. Practical Implications

While our hypothesis was not confirmed, the information provided by this study has important implications in understanding how WASH interacts with other variables and disparities to influence children’s health [34]. This is essential for future programming that aims to address WaSH in low- and middle-income (LMIC) countries that have low levels of WaSH access. Our findings emphasize that children’s health outcomes are multifaceted and require a holistic approach that begins with caregivers. This can influence policy and programming related to global efforts in LMIC countries to address WaSH by integrating education and interventions at the household level, not just for the mother or children.

## Figures and Tables

**Figure 1 ijerph-22-00706-f001:**
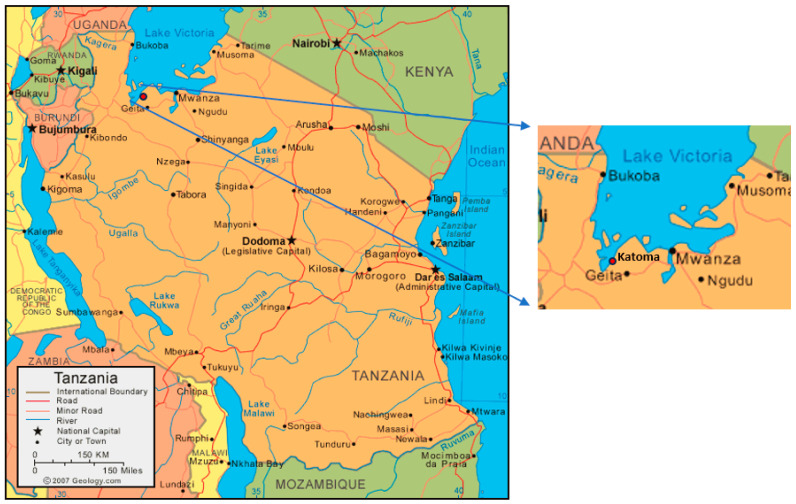
A map of Tanzania detailing the location of Katoma in Geita, Tanzania [16].

**Figure 2 ijerph-22-00706-f002:**
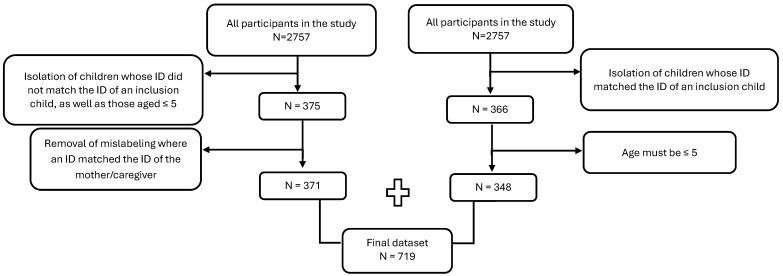
Diagram of inclusion criteria for final dataset used for analysis.

**Figure 3 ijerph-22-00706-f003:**
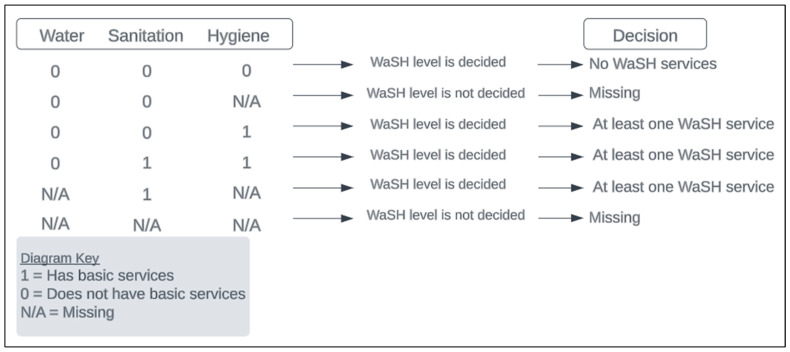
Diagram of examples showing how missingness was handled when categorizing WaSH for the final models.

**Table 1 ijerph-22-00706-t001:** Demographic characteristics of the 719 children stratified by WaSH status.

Characteristics	No Basic WaSH ^a^ Status ^b^ n = 419	At Least one Basic WaSH Status n = 204
Sick ^c^, (Yes), N (%)	95 (22.67)	42 (20.59)
Education of the mother ^d^ (completed primary), N (%)	185 (44.15)	100 (49.02)
Vaccination ^e^ (Yes), N (%)	345 (82.34)	163 (79.90)
Food insecurity ^f^ (Moderate or severe), N (%)	214 (51.07)	65 (31.86)
Income ^g^ (Income-generating), N (%)	64 (15.27)	21(10.29)
Wealth ^h^, N (%)		
Poor	202 (48.21)	39 (19.12)
Middle	140 (33.41)	90 (44.12)
Rich	57 (13.60)	62 (30.39)
Sex, Male, N (%)	211 (50.36)	108 (52.94)
Hamlet ^i^, N (%)		
Katoma Center	119 (28.40)	58(28.43)
Mataho	91 (21.71)	36 (17.65)
Chang’ombe	143 (34.13)	74 (36.27)
isoji	44 (10.50)	27 (13.24)
Ruchiri	22 (5.25)	9 (4.41)

a. WaSH = water access, sanitation, and hygiene. b. Whether the household met the basic requirements for water access, sanitation, or hygiene. c. Whether or not the child reported having had diarrhea or a fever in the past month. d. The education level of the mother/caregiver of the inclusion child within the household (completed primary education or not). e. Whether the inclusion child within the household had a vaccination card (yes or no). f. The food insecurity level of the household (mild/no food insecurity or moderate/severe food insecurity). g. Usage of animal products and common property resources (sustenance or income-generating). h. Wealth is the total value of the household assets. i. Hamlet = small community of people.

**Table 2 ijerph-22-00706-t002:** Household factors and their association with children under five having diarrhea.

Characteristics	Model I	Model II	Model III	Model IV
	OR (95% CI) *p*-Value	OR (95% CI) *p*-Value	OR (95% CI) *p*-Value	OR (95% CI) *p*-Value
**WaSH**				
No basic services	Reference	Reference	Reference	Reference
At least one basic service	2.04 (1.05, 3.98) 0.036 *	1.34 (0.68, 2.67) 0.397	1.5 (0.93, 2.64) 0.090	1.9 (0.66, 2.166) 0.558
**Food insecurity**				
Mild or none		1.25 (0.61, 2.59) 0.538		1.32 (0.73, 2.37) 0.354
Moderate or severe		Reference		Reference
**Education of the mother**				
Did not complete primary		Reference		Reference
Completed primary		1.72 (0.91, 3.25) 0.095		1.54 (0.94, 2.52) 0.089
**Vaccination card**				
No		Reference		Reference
Yes		0.67 (0.30, 1.50) 0.331		0.46 (0.20, 1.04) 0.063
**Income**				
Sustenance		Reference		Reference
Income-generating		1.13 (0.49, 2.65) 0.771		1.00 (0.50, 1.99) 0.997
**Hamlet**				
Katoma Center		Reference		Reference
Mataho		0.41 (0.17, 1.01) 0.053		0.51 (0.26, 1.01) 0.053
Chang’ombe		0.51 (0.24, 1.10) 0.084		0.60 (0.33, 1.08) 0.083
Kisoji		0.64 (0.23, 1.84) 0.411		0.94 (0.40, 2.22) 0.889
Ruchiri		0.07 (0.01, 0.65) 0.012 *		0.09 (0.02, 0.37) 0.001 *

Note: Model I—the null logistic regression model of inclusion children with WaSH being the only explanatory variable. Model II—the same logistic regression model of inclusion children, but adjusting for food insecurity, education of the mother, vaccination status, income, and hamlet. Model III—the null mixed-effects model of all children with a random effect for households where WaSH was the only explanatory variable. Model IV—the same mixed-effects model of all children, but adjusting for the same covariates as Model II. * Significant at the α = 0.05 level.

**Table 3 ijerph-22-00706-t003:** Household factors, including wealth, and their association with children under five having diarrhea.

Characteristics	Model II	Model IV
	OR (95% CI), *p*-Value	OR (95% CI), *p*-Value
**WaSH**		
No basic services	Reference	Reference
At least one basic service	1.46 (0.70, 3. 04), 0.306	1.13 (0.62, 2.06), 0.696
**Food insecurity**		
Mild or none	1.16 (0.52, 2.59), 0.708	1.21 (0.66, 2.21), 0.543
Moderate or severe	Reference	Reference
**Education of the mother**		
Did not complete primary	Reference	Reference
Completed primary	1.58 (0.80, 3.10), 0.187	1.34 (0.79, 2.27), 0.281
**Vaccination card**		
No	Reference	Reference
Yes	0.70 (0.30, 1.61), 0.398	0.50 (0.22, 1.12), 0.089
**Wealth**		
Poor	Reference	Reference
Middle	1.64 (0.75, 3.59), 0.218	1.80 (1.00, 3.25), 0.051
Rich	1.05 (0.40, 2.77), 0.916	1.24 (0.60, 2.60), 0.560
**Hamlet**		
Katoma Center	Reference	Reference
Mataho	0.38 (0.15, 0.96), 0.041 *	0.50 (0.25, 1.00), 0.049 *
Chang’ombe	0.49 (0.22, 1.07), 0.074	0.60 (0.33, 1.09), 0.094
Kisoji	0.71 (0.23, 2.19), 0.547	1.23 (0.47, 3.17), 0.673
Ruchiri	0.06 (0.01, 0.71), 0.026 *	0.08 (0.02, 0.34), 0.001 *

Note: Model II—the logistic regression model of inclusion children, adjusting for food insecurity, education of the mother, vaccination status, hamlet, and wealth replacing income; Model IV—the mixed-effects model of all children with a random effect for households adjusting for the same covariates as Model II. * Significant at the α = 0.05 level.

**Table 4 ijerph-22-00706-t004:** Household factors, redefining WaSH, and their association with children under five having diarrhea.

Characteristics	Model II	Model IV
	OR (95% CI), *p*-Value	OR (95% CI), *p*-Value
**WaSH**		
No basic services	Reference	Reference
At least one basic service	121 (0.56, 260), 0.631	0.94 (0.48, 1.82), 0.850
At least two basic services	1.53 (0.45, 5.16), 0.493	2.52 (0.71, 8.97) 0.512
**Food insecurity**		
Mild or none	2.42 (1.03, 5.67), 0.042 *	1.99 (0.97, 4.09), 0.060
Moderate or severe	Reference	Reference
**Education of the mother**		
Did not complete primary	Reference	Reference
Completed primary	3.26 (1.29, 8.25), 0.013 *	2.24 (1.10, 4.59), 0.027 *
**Vaccination card**		
No	Reference	Reference
Yes	0.53 (0.22, 1.28), 0.159	0.30 (0.10, 0.92), 0.036 *
**Income**		
Sustenance	Reference	Reference
Income-generating	1.57 (0.59, 4.19), 0.363	1.16 (0.57, 2.36), 0.683
**Hamlet**		
Katoma Center	Reference	Reference
Mataho	1.83 (0.48, 7.05), 0.376	1.38 (0.44, 4.30), 0.581
Chang’ombe	1.33 (0.37, 4.85), 0.662	1.29 (0.40, 4.09), 0.669
Kisoji	1.75 (0.48, 6.33), 0.391	2.24 (0.76, 6.58), 0.143
Ruchiri	0.21 (0.02, 2.59), 0.224	0.23 (0.04, 1.22), 0.085

Note: Model II—the logistic regression model of inclusion children, adjusting for food insecurity, mother’s education, vaccination status, income, and hamlet. WaSH was no longer binary, and a new sampling weight was used to account for the low representation of children with at least two services. Model IV—the mixed-effects model of all children with a random effect for households adjusting for the same covariates as Model II. The new sampling weight and definition of WaSH were used. * Significant at the α = 0.05 level.

## Data Availability

The data presented in this study are available on request from the corresponding author.

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
