# Peer review of "Empowering Women, Enhancing Health: The Role of Education in Water, Sanitation, and Hygiene (WaSH) and Child Health Outcomes"

_ijerph, 2025, doi:10.3390/ijerph22050706_

Round 1
Reviewer 1 Report
Comments and Suggestions for Authors
The manuscript with a somewhat "eye-catching" title examines an important topic, namely whether the existence or lack of a satisfactory WASH service plays a role in diarrheal diseases among children under 5 years of age in a LIC (Tanzania).
According to the results, WASH is not alone, but in association with other determinants (mother's education, poverty, etc.) is already significantly related to health outcomes.
The research deserves attention not only because of the large number of cases, but also because of the accurate methodology. At the same time, the symptoms established for lay reports mean uncertainty, although it must be accepted that better identification is not possible due to the state of the healthcare system.
The value of the study is that it reviews the limitations of the studies, and also that it suggests the need for a holistic approach, which should also have consequences for health policy.
Author Response
We sincerely appreciate the time and effort you dedicated to reviewing our manuscript. Your insightful comments have greatly contributed to strengthening the quality of our work. We have carefully incorporated your comments into our revisions. Since no revisions were requested, we only incorporated the comments.

Reviewer 2 Report
Comments and Suggestions for Authors
Thank you for bringing out an important issue to research on "Empowering Women, Enhancing Health: The Role of Education in WaSH and Child Health Outcomes". It's timely and interesting research. However, to make it for more readable to large audience some of my suggestions are:
Abstract:
Abstract needs to improve. Rather than highlighting, please improve the methods and results section.
Introduction:
I would request the authors to put additional literature related to WASH. There is significant amount of literature are available, and few more relevant literatures can be added. Also, at the end of introduction, include the outline of the paper.
Methods:
Is it possible to include the lat long of the households surveyed in the map. This will certainly improve the quality of the map.
Can inclusion and exclusion criteria have explained in the text. its missing as of now.
Statistical methods are not telling clearly. For example, what exactly methods are used by the authors are not clear. Please spend some time on it before resubmission the manuscript.
Results:
Segregate results into sub-section. After methodology get changed, there is some expected changes into the results and discussion section and in the conclusion too.
I wish the author's good luck.
Author Response
We sincerely appreciate the time and effort you dedicated to reviewing our manuscript. Your insightful comments have greatly contributed to strengthening the quality of our work. We have carefully addressed all of your suggestions, and where we were unable to make the recommended changes, we have provided clear justifications.
Following the guidelines of the journal, all edits have been highlighted in the manuscript Word doc.

Reviewer 3 Report
Comments and Suggestions for Authors
Thank you very much for the opportunity to review this manuscript. Overall, it is logically structured and well-written. It explores a highly relevant public health issue that warrants urgent and effective action to save human lives. I congratulate you on successfully conducting an important research study that will raise awareness about the challenges faced by vulnerable communities.
Please allow me to share my feedback, which is intended to improve the overall clarity and understandability of the manuscript.
First, I would like you to please consider some general points, listed below but also marked in the manuscript:
- Please ensure that there are no grammar errors. For example, the endings of verbs depend on whether the subject is singular or plural.
- Please report if you applied and received approval from the ethics review committee.
- Please report if you asked for and received written/verbal informed consent from study participants before collecting their data.
- Whenever you report a statistic, please ensure you include a reference (with author and publication year).
- Please create a table of operational definitions related to WaSH adequacy. Please include references, i.e. which organization/agency provided the definitions.
- Please report the (95%) confidence interval (CI).
- The Results section is too long and difficult to follow. It seems you have repeated many things that are already shown in the tables. To improve readability, I suggest breaking down the Results section into different parts and highlighting only the findings that you found to be interesting or unexpected. Please see manuscript for specific comments.
- Thank you for reporting the study limitations. However, it seems that you may have missed a few limitations. Please see manuscript for specific comments.
Second, I would like you to please consider some specific points, which I have highlighted in the manuscript. Please see my comments in the manuscript (PDF) for more information.
Finally, thank you once again for undertaking this research and writing this manuscript. Once you have addressed my feedback, I think your manuscript will be much improved and ready for another review. Thank you very much for your consideration.

Overall the quality of English language is good. However, there are grammatical errors. I have marked those in the manuscript (PDF) for the authors to correct.
Author Response
We sincerely appreciate the time and effort you dedicated to reviewing our manuscript. Your insightful comments have greatly contributed to strengthening the quality of our work. We have carefully addressed all of your suggestions, and where we were unable to make the recommended changes, we have provided clear justifications.
Following the guidelines of the journal, all edits have been highlighted in the Word document.

Reviewer 4 Report
Comments and Suggestions for Authors
Dear authors, congratulations for your work, it shows effort. I am going to make some recommendations to help you with your article. Limitations should be discussed. Add a section on future research and another on practical applications where each of them is explained in detail and where the transfer to society of the article can be seen. On the other hand, in methodology I have missed a section on the ethics of your study and how you have carried out this whole process from the costs to the anomization, etc. Add the hypotheses of your study and articulate the discussion with these. Lots of encouragement, you are doing a great job.
Author Response

(The authors gave the same response as above.)

Round 2
Reviewer 4 Report
Comments and Suggestions for Authors
The authors have adequately addressed and implemented all the requested changes. The revised manuscript reflects careful attention to the feedback provided, and the improvements are evident across all sections. The bibliographic references are appropriate, specific, and well integrated into the theoretical framework.
The study effectively meets its stated research objectives. The scientific method employed is rigorous, precise, and clearly articulated. Furthermore, the research complies with the ethical principles outlined in the Declaration of Helsinki, demonstrating a strong commitment to ethical standards in research involving human participants.
The discussion section is well-written, logically structured, and provides a thoughtful interpretation of the findings. It successfully integrates the results with relevant literature and theoretical frameworks, offering valuable insights for both academic and clinical audiences.
In conclusion, I commend the authors for their thorough and high-quality work. I encourage them to continue their research in this important field, as their contributions are both timely and impactful.